# Rice Compounds with Impact on Diabetes Control

**DOI:** 10.3390/foods10091992

**Published:** 2021-08-25

**Authors:** Cristiana Pereira, Vanda M. Lourenço, Regina Menezes, Carla Brites

**Affiliations:** 1National Institute for Agricultural and Veterinary Research (INIAV), I.P., Av. da República, 2780-157 Oeiras, Portugal; cristianapereirags@gmail.com; 2Faculty of Sciences and Technology (FCT) & Center for Mathematics and Applications (CMA), NOVA University of Lisbon, 2829-516 Caparica, Portugal; vmml@fct.unl.pt; 3CBIOS—Universidade Lusófona’s Research Center for Biosciences & Health Technologies, Campo Grande 376, 1749-024 Lisboa, Portugal; regina.menezes@ulusofona.pt; 4CEDOC, Chronic Diseases Research Centre, NOVA Medical School|Faculdade de Ciências Médicas, Universidade NOVA de Lisboa, Campo dos Mártires da Pátria, 130, 1169-056 Lisboa, Portugal; 5iBET, Instituto de Biologia Experimental e Tecnológica, Apartado 12, 2781-901 Oeiras, Portugal; 6GREEN-IT Bioresources for Sustainability, ITQB NOVA, Av. da República, 2780-157 Oeiras, Portugal

**Keywords:** diabetes, rice bran, phytic acid, vitamin E, γ-oryzanol, γ-aminobutyric acid, ferulic acid, GLUT1, SGLT2, IAPP

## Abstract

Rice is one of the most cultivated and consumed cereals worldwide. It is composed of starch, which is an important source of diet energy, hypoallergenic proteins, and other bioactive compounds with known nutritional functionalities. Noteworthy is that the rice bran (outer layer of rice grains), a side-stream product of the rice milling process, has a higher content of bioactive compounds than white rice (polished rice grains). Bran functional ingredients such as γ-oryzanol, phytic acid, ferulic acid, γ-aminobutyric acid, tocopherols, and tocotrienols (vitamin E) have been linked to several health benefits. In this study, we reviewed the effects of rice glycemic index, macronutrients, and bioactive compounds on the pathological mechanisms associated with diabetes, identifying the rice compounds potentially exerting protective activities towards disease control. The effects of starch, proteins, and bran bioactive compounds for diabetic control were reviewed and provide important insights about the nutritional quality of rice-based foods.

## 1. Introduction

Diabetes Mellitus (DM) is a spectrum of metabolic diseases affecting hundreds of millions of people worldwide, whose numbers are expected to continue increasing in the following years [1]. As a diet-related disease, DM prevention requires a strict diet along with therapeutic control [2]. The disease is characterized by increased blood glucose levels, which can arise from two main causes, leading to its classification as type 1 and type 2 Diabetes Mellitus (T1DM and T2DM, respectively) [3,4]. In T1DM, an autoimmune disease corresponding to ~5–10% of all diabetes cases, the immune system destroys the insulin-producing β-cells of the pancreas. In T2DM (hereafter referred to as diabetes), a condition that corresponds to ~90–95% of all diabetes cases [5], cells become resistant to insulin, leading to the dysregulation of glucose metabolization.

Rice, one of the staple foods of the world’s population, is consumed as white cooked polished grain and is a high glycemic index (GI) food (64–93%) [6]. Given its fundamental importance in the human diet, it is imperative to exhaustively evaluate rice nutritional composition, which can differ among varieties. Evaluation of the nutritional composition of white rice indicates that it is essentially composed of starch, with rice protein being the second largest nutrient in white rice (Table 1). These nutrients and other compounds present in the brown rice grain are relevant in the control of metabolic diseases, especially in diabetes control.

Brown rice has a higher content of nutritional compounds compared with white rice, except starch (Table 1). The health benefits of brown rice are mainly due to their proteins, lipids, vitamins, minerals, and dietary fiber contents [7,8]. Several in vivo studies have shown the benefits of brown rice-based or brown rice-supplemented diets in diabetes control, especially in reducing the glycemic index [9,10,11].

The rice milling industry generates large quantities of side-stream products without great valorization, such as rice bran [12], which represents about 10–13% of brown rice grain [13]. Notably, rice bran has greater lipids and dietary fibers than white rice (Table 1).

Rice bran contains less carbohydrates than white rice and twice as much protein (Table 1), and it is therefore an important protein source (such as rice albumin). Remarkably, rice bran is also rich in bioactive compounds such as γ-oryzanol, ferulic, phytic and γ-aminobutyric acids, tocopherols, and tocotrienols (vitamin E) (Table 2), which may differ in concentration depending on the variety and type of rice [14]. These compounds have been shown to exert hypolipidemic and anti-hypertensive effects, in addition to contributing to the mitigation of oxidative stress [14,15,16,17], a pathological process associated with diabetes [18,19]. Rice bran thus represents a great nutritional source of lipids, proteins, and non-starchy polysaccharides (accounted in the crude fiber fraction), richer than in white rice (Table 1), and can be used to produce new food ingredients such as rice bran oil or protein isolates.

The main purpose of this review is to describe the potential therapeutic effects of rice compounds in diabetes, especially the relevant macronutrients (starch, proteins; Section 2) via the control of the glycemic index and their effects in glycemia, and to discuss the value of rice bran compounds (Section 3). The effects of rice glycemic index, macronutrients, and bioactive compounds on the pathological mechanisms associated with diabetes are discussed to support the exploitation of rice compounds as functional ingredients in diabetes control (Section 4).

## 2. Rice Macronutrients

### 2.1. Starch

Starch is the main nutrient present in rice (77.6%) (Table 1) being, for this reason, the most influential ingredient in rice quality definition. It is also the main provider of the greatest amount of diet energy. Starch can be classified into three main categories based on its in vitro digestion: rapid digesting starch (RDS), slow digesting starch (SDS), and resistant starch (RS) [38]. Resistant starch is defined as starch, or products derived from starch digestion, that are not digested in the small intestine. It is important to address its effect on blood glucose as starches from different rice processed foods can have different digestion and absorption rates. Some studies have indicated that variability in the starch digestibility was associated with the great variability of the glycemic index (GI) in different types of rice [38,39], highlighting the importance of selecting new lower GI-varieties (Table 3) [40].

As such, the use of foods rich (or enriched) in RS is an asset and a potential strategy for reducing starch digestion rate [41,42,43] and blood glucose levels [38,44,45]. RS can be divided into five types: RS1 is a physically inaccessible starch, presented in compact food matrices, present essentially in whole grains; RS2 comes from starch that has not been gelatinized, appearing in non-gelatinized foods; RS3 exists in starch after retrogradation, showing better functional characteristics, namely low digestibility; RS4 is a modified starch prepared through enzymatic, chemical, physical, or genetic modifications, managing to be artificially increased; and RS5 is formed by an amylose-lipid complex [46,47]. Currently, there is a growing interest in RS4 for possible modification and improvement through physical, chemical, and enzymatic methods in order to obtain different types of RS with greater applicability [47,48]. Rice RS content can vary between 0.1–3.2% [49,50,51,52]. The properties of rice starch, such as size granules and crystallinity, amylose/amylopectin ratio, and gelatinization characteristics can also be predictors of starch digestion rate [38,39,51,52,53]. Extrinsic factors such as crop edaphoclimatic conditions largely influence basic starch composition, especially proteins and lipids, which may reduce the access of digestive enzymes to the starch hydrolysis [45,54,55]. These factors also influence the concentration of rice bioactive compounds [51,56,57], which will be discussed later.

Rice amylose content is the main starch feature that influences starch digestion. Its linear structure hampers amylolytic enzyme action. It is more resistant to digestion than amylopectin, with a branched structure. Therefore, high-amylose rice varieties were reported to have slower digestion starch than low-amylose varieties [39,58,59]. The positive correlation between amylose and RS concentrations in samples with a low GI is also reported [51,52]. In addition to chemical, physical, and enzymatic methods for modifying RS, genetic modification has also become important. Genetic manipulation acts essentially by starch manipulation, aiming to create highly resistant starch varieties. For this purpose, the genetic starch chain is modified through mutagenesis (with starch biosynthesis related genes) to accumulate a higher amylose content [47,60].

### 2.2. Proteins

White rice contains about 4.5–10.5% (Table 1) protein, being its second highest nutrient. Rice bran, with 11.3–14.9% total protein, contains twice the amount of protein than white rice. According to some authors, rice proteins play an important role in the starch digestion rate when interacting through their connections with the starch granules [53,55]. Ye et al. (2018) showed an increase in starch digestion after the removal of total intrinsic protein from rice flour. This suggests that varieties with higher protein content may be beneficial in reducing GI. 

There are four types of proteins present in the rice grain: glutelin (60–80%), albumin (4–22%), globulin (5–13%), and prolamin (1–5%). Rice bran contains albumin (24–37%), globulins (15–36%), glutelins (11–38%), and prolamins (2–5%) [61].

In addition to these reports, rice proteins stand out for their health benefits, namely because they are hypoallergenic (gluten free), being widely used in the formulation of baby foods and functional drinks, especially due to their functionality and the fact that they are amenable for improvement by enzymatic hydrolysis [62,63]. Albumin, referred to as a water-soluble protein, having easy digestion and absorption [61,62], stands out in this context. The albumin in rice grain is essentially present in rice bran (3.2–4.9 g/100 g) [64]. It has been highlighted in some studies for its inhibitory effect on starch digesting enzymes (α-amylase), thus contributing to decreasing the foods GI [65]. The rice albumins, with a molecular weight of 14–20 kDa, are recognized as an α-amylase inhibitor [7,65].

### 2.3. Lipids

Rice bran is a greater source of lipids (15.0–19.7%) than white rice (Table 1). There are two types of lipids present in rice grain: starch lipids, present inside starch granules, and non-starch lipids, outside starch granules [7]. Triacylglycerols are the major non-starch lipids present in the rice grain; free fatty acid and lysophospholipids are the major starch lipids in the rice grain [43]. Starch lipids, although they exist in a smaller proportion, still play an important role in starch functionality and quality, affecting its digestibility by modifying starch swelling, gelatinization, and pasting [66]. These complexes are formed during the synthesis of starch. They are best known for forming complexes with amylose molecules stabilized by hydrogen bounds [43,66], which inhibit the swelling of rice and reduce susceptibility to starch digestive enzymes. The study by Ye et al. (2018) shows that the presence of endogenous lipids has effects on rice flour digestibility. There was an increase in starch digestibility after the removal of the endogenous lipids, confirming the starch-lipid complex interactions [55]. In addition to the importance on starch digestibility, the lipid fraction of the rice grain also contains bioactive compounds (detailed more in Section 3 and Section 4) such as ϒ-oryzanol, tocopherols and tocotrienols, ferulic acid, and triterpene alcohols, with antioxidant properties and effects on diabetes control [67,68,69].

### 2.4. Dietary Fiber

Dietary fibers are known as non-digestible carbohydrates and are commonly classified according to their solubility in water, viscosity, or fermentation rate [70]. In soluble fibers, pectin and inulin stand out, whereas, in insoluble fibers, RS, oligosaccharides, cellulose, and hemicellulose are generally predominant [70]. The consumption of dietary fiber has health benefits, namely effects on the control of diseases such as diabetes and hyperlipidemia [43,69,71,72].

Rice bran is a great source of dietary fiber (19–29%), compared with white rice (Table 1). The most common fibers in rice are cellulose, arabinoxylans, and pectic substances [43]. Dietary fiber in rice can be bound with other components, the soluble fibers are complex, with proteins, lipids, and other compounds. Insoluble fibers are not normally easily digested and are fermented in the large intestine, playing an important role. They can confer different rheological properties that influence the quality, sensory, and functional properties [16], and can be used in the development of functional food products. A study by Qureshi et al. (2002) shows improvements in the serum glucose content after consumption of rice bran fiber [71]. Chang et al. (2014) found that there is a correlation between dietary fiber and the glycemic index. Most et al. (2005) reports beneficial effects after the consumption of rice fiber in pos-prandial lipidemia levels [73]. 

## 3. Rice Bran Compounds

### 3.1. γ-Oryzanol

γ-Oryzanol is found mainly in the lipid portion of the rice bran at a concentration ranging from 59.4 to 912.0 mg/100 g [21,22,23,24], depending on factors such as the extraction method and rice bran origin. γ-oryzanol is a mixture of ferulate esters (Figure 1), which are formed by esterification of the hydroxyl group of sterols (campesterol, stigmasterol, β-sitosterol) or triterpene alcohols (cycloartanol, cycloartenol, 24-methylenecycloartanol, cyclobranol) with a carboxylic group of ferulic acids [74,75]. Despite that, almost 95% total γ-oryzanol content is essentially composed of four principal ferulate compounds: 24-methylenecycloartanyl ferulate, cycloartenyl ferulate, campesteryl ferulate, and β-sitosteryl [74]. Only a few studies have addressed γ-oryzanol bioavailability. According to Kobayashi et al. (2020), γ-oryzanol can be absorbed in its form or it gives rise to ferulic acid and phytosterols metabolites.

### 3.2. Phytic Acid

Phytic acid (PA), also known as phosphorus phytate or myo-inositol-1,2,3,4,5,6-hexaphosphate, is an organic acid essentially found in seeds and cereal brans, from where it can be extracted. It represents about 65 to 73% of the total phosphorus content in rice. In the bran, PA is present in greater abundance (90%), varying from 4.0 to 22.5 g/100 g [25,26,28,29]. As for its bioavailability, Schlemmer et al. (2001) demonstrated that, after ingestion, up to about 66% of PA is degraded and absorbed in the stomach and intestine [76]. This process can be accelerated by the presence of phytase enzymes present in the diet or microbial phytases in the large intestine. PA dephosphorylation is metabolized to various inositol phosphate derivatives, which are more easily absorbed [76,77]. 

### 3.3. Ferulic Acid

The most abundant phenolic compound found in the endosperm, bran, and whole grain of rice is feruled acid (FA), corresponding to 56–77% of total phenolic acids. The main function of FA is providing rigidity to the cell wall, with effects on the germination of seeds. FA is composed of an aromatic ring to which a carboxyl group is attached (Figure 1). Also known as 3-methoxy 4-hydroxycinnamic acid, it is present in a concentration range of 1.4–225.4 mg/100 g in rice bran [31]. Owing to its easy absorption and metabolization by the organism, FA derived from rice bran is considered a compound with great nutraceutical potential [17]. 

### 3.4. γ-Aminobutyric Acid (GABA)

γ-Aminobutyric acid (GABA) is produced by the decarboxylation of L-glutamic acid by glutamate decarboxylase during the germination process of whole rice, and it is also found in other plants and cereals [78,79]. GABA consists of a chain of four-carbon non-protein with a deprotonated carboxyl group and the protonated amino group at the other end of the chain [80] (Figure 1). Rice bran contains a large amount of glutamic acid, allowing the use of this by-product in the synthesis of GABA by fermentation [34,81,82]. GABA is found essentially in whole rice and rice bran, and its content can be increased with the germination process of the grain [83]. The amount of GABA in rice bran ranges from 10.7–58.0 mg/100 g (before germination) and 90.0–350.0 mg/100 g (after 10–12 h of rice bran germination) [33,34,35]. Thus, increased levels of GABA can be obtained by reinforcing the germination status of the grain [83,84,85].

### 3.5. Tocopherols and Tocotrienols (Vitamin E)

Vegetable oils are common sources of tocopherols and tocotrienols (vitamin E), some of which contain predominantly tocopherols (wheat oil), while others contain mainly tocotrienols (rice bran oil) [36,86]. Tocopherols and tocotrienols are a group of antioxidants [36,87,88] present in the unsaponifiable fraction of rice bran oil. They exist in greater quantities in the bran (17.0–22.9 mg/100 g) than in the other fractions of rice [20,36,37]. Vitamin E is composed of a chromanol ring with a carbon side chain that gives it its antioxidant properties by reacting with peroxyl radicals. It is composed of eight different stereoisomers α-, β-, γ-, and δ-tocopherols, with a phytyl side chain, and α-, β-, γ-, and δ-tocotrienols, with an unsaturated isoprenyl side chain (with three double bonds). The biological role of tocopherols and tocotrienols is different depending on their homologous forms, which are distinguished by the number and location of the methyl groups in the chromatol ring [87,88].

## 4. Metabolic Mechanisms and Bioactive towards Diabetes

### 4.1. Glycemic Index of Rice and Effects in Glycemia

Post-prandial glycemia is defined as the blood glucose concentration after a meal and is largely influenced by the GI of foods containing carbohydrates [57,89]. GI is a measure of the glucose response to the intake of a fixed amount of carbohydrates available in the food [57,89] and can be determined in vivo [90]. Determination of GI in vivo is performed in accordance with the standard GI testing protocol, in which foods are tested on fasting individuals, with a reference food (with known glucose concentration). Before and over the time of food digestion, several blood samples are collected, which allows the determination of blood glucose concentration, thus allowing estimation of the food GI. GI can be predicted by in vitro methods assessing the rate of starch hydrolysis [91]. A reference food is also used in in the vitro method, but the process of digestion is done in vitro using a buffer solution containing the sample food and digestive enzymes that hydrolyses complex carbohydrates in sugars (α-amylase and the α-glucosidase). Throughout the digestion process, the glucose formed is determined, which makes it possible to estimate the expected GI.

GI is directly related to starch digestibility and absorption of the compounds generated after starch digestion, being therefore related to factors affecting digestibility or absorption. The rapid- or slow-digestion nature of cooked starch-containing foods has a significant impact on blood glucose concentration [92]. SDS and RS foods, which are indicated by starch being digested gradually or not hydrolysed, are characterized as low GI foods [45]. The presence of bioactive compounds, mainly in the rice bran, can also contribute to the reduction of GI and the valorization of its side-stream product and incorporation into new products. Most studies related to the effect of rice compound intake on the post-prandial glucose decrease were performed using several models (Table 4). It is important to note that these studies evaluated the effects of pure forms of rice bioactive compounds or as supplements in other foods or meals, and they were mostly carried out on animals. Thus, the potential benefits for human health should be extrapolated with caution.

### 4.2. α-Amylase and α-Glucosidase Inhibition

Inhibition of α-glucosidase and α-amylase is another strategy to control hyperglycemia [103]. Complex carbohydrates reaching the small intestine cannot be directly absorbed by enterocytes of the epithelium; they need to be first degraded by the glycohydrolases α-amylase and α-glucosidase. Inhibitors of these enzymes delay the increase of glucose concentration in the small intestine and efficiently promotes apical glucose transporters (GLUT2 and SGLT1) insertion, preventing the increase of blood glucose levels [104]. Compounds inhibiting these enzymes may contribute to the reduction of GI of carbohydrate-containing foods, allowing the design of healthy products for diabetes control.

α-Amylases (EC 3.2.1.1) are enzymes responsible for the hydrolysis of the α-(1,4)-glycosidic bonds of starch forming dextrins. There are two types of α-amylases present in mammals. The salivary α-amylase is active during chewing in the mouth and acts at a higher pH, being inactive in the stomach. The pancreatic α-amylase acts in the first stage of carbohydrates digestion in the small intestine [105].

α-Glucosidases (EC 3.2.1.20) are a group of enzymes responsible for the hydrolysis of non-reducing terminal α-(1,4)-glycosidic bonds of oligosaccharides and polysaccharides [65] in monosaccharides such as glucose. These enzymes, including trehalase, lactase, maltase-glucoamylase complex, and sucrase-isomaltase complex, are bound to the wall/edge of the small intestine and degrade the dextrins resulting from the digestion of amylase before the absorption of glucose by the intestine wall. The studies addressing the effect of rice compounds on α-amylase and α-glucosidase activities are summarized in Table 5.

### 4.3. Modulation of Glucose Transporters 

Glucose transport is one of the main nutrient uptake systems in intestinal cells. In fact, intestinal glucose transport and absorption are fundamental processes to understand postprandial fluctuations in blood glucose concentration with implications in diabetes control. It is catalyzed by a group of glucose transporters (GLUT) and sodium-coupled glucose co-transporters (SGLT) [109]. The GLUT family of transporter proteins is composed of GLUT1-12,14, GLUT1 to 4 being the most studied. In addition, there is the myo-inositol transporter (HMIT). SGLTs and GLUTs are responsible for the transport of monosaccharides through the lipid bilayer of the plasma membrane in eukaryotic cells. These transporters in the intestine can be classified according to their affinity and capacity to transport glucose into blood (GLUT2) and enterocytes (SGLT1 and GLUT2) [110]. Although transport by GLUT2 is facilitated, transport by SGLT1 is active via the Na/K-pump, which provides energy and maintains the sugar gradient [109]. 

Before a meal, when the sugar concentration in the lumen and blood is low, the only active transporter is SGLT1, with GLUT2 being present in a very low concentration. After a meal, when the glucose concentration in the lumen is high, GLUT2 is activated in the apical enterocyte membrane [110], increasing glucose uptake ~3-fold. Before absorption in the small intestine by enterocytes, carbohydrates are digested in monosaccharides that are transported by SGLT1, GLUT2, and GLUT5 (for fructose) [109]. GLUT2 is responsible for 75% of total glucose absorption [104]. Thus, activation of GLUT2 is affected by the sugar content of the diet. In diabetes, there is an increase in intestinal glucose absorption, due in large part to the overexpression of GLUT2 and SGLT1 transporters [104]. Thus, diet supplements inhibiting glucose absorption through SGLT1 and GLUT2 can potentially be used for blood glucose regulation and insulin secretion therapies [104,110,111,112]. Table 6 summarizes in vivo and in vitro studies supporting the modulation of GLUT2 and SGLT1 expression by rice compounds. 

### 4.4. Other Molecular Mechanisms Associated with Diabetes

Diabetes is a multi-factorial disorder and, in addition to the mechanisms mentioned above, other molecular pathways contribute to disease onset and progression, in particular those related to pancreatic β-cell dysfunction [116,117,118]. The intracellular aggregation of Islet amyloid polypeptide (IAPP) and the accumulation of extracellular amyloid deposits in the vicinity of pancreatic β-cells, which have been found in over 95% of T2DM patients [119], are some of these pathways. In response to glucose stimulus, β-cells secrete insulin and IAPP, or amylin. IAPP is a highly amyloidogenic hormone that, under physiological conditions, contributes to regulating glucose levels by inhibiting insulin and glucagon secretion, also controlling adiposity and satiation [120]. In T2DM, cells become resistant to insulin, leading to the dysregulation of glucose metabolism [121]. As a compensatory mechanism, there is an increase of β-cell activity to produce more insulin. Increased activity of β-cells leads to the overproduction of IAPP, which in turn triggers IAPP intracellular oligomerization and the accumulation of extracellular amyloid plaques [122,123].

The impact of IAPP intracellular oligomerization affecting fundamental β-cell functions [124], the deposition of IAPP fibrils between β-cells and capillary endothelial cells, likely impairs glucose and nutrients flow, favoring cell death and interfering with insulin secretion [124,125]. In addition, the high demand for insulin/IAPP production by β-cells trigger endoplasmic reticulum (ER) stress causing progressive dysfunction and ultimately the death of these cells [118]. Thus, strategies aimed at improving function or decreasing the death of β-cells are currently under investigation as a means to delay disease onset and progression. Table 7 summarizes the in vivo and in vitro studies, pointing out the potential of rice bioactivities as protective compounds against these pathological mechanisms. 

### 4.5. Highlight on Rice Major Bioactive Compounds towards Diabetes

#### 4.5.1. γ-Oryzanol

γ-oryzanol exerts a plethora of bioactivities, antioxidant and hypocholesterolemic properties especially, and it has been mainly explored in terms of cosmetics, pharmaceuticals, and as food supplements [16,125]. The γ-oryzanol activity related to diabetes is associated with its antioxidant and hypocholesteolemic properties [18,126]. The latter comes from its ability to inhibit the hydrolytic function of the enzyme cholesterol esterase (CEase), essential to produce free cholesterol in the lumen. As for its potential antioxidant activity, the component of γ-oryzanol standing out with the greatest antioxidant activity in vitro is 24-methylenecycloartanyl [74,127,128], due to the presence of a hydroxyl group on the phenolic ring [129]. It remains to be seen whether γ-oryzanol concentrations reaching internal tissues are sufficient to exert this potential activity. γ-oryzanol antioxidant properties are also mediated through the activation of the antioxidant enzymes, including superoxide dismutase, catalase, and glutathione peroxidase, responsible for the neutralization of reactive oxidative species (ROS) [126], as determined by in vivo studies in mice [18,130,131,132,133]. In fact, the administration of γ-oryzanol reduces blood glucose levels as well as the symptoms of diabetic neuropathy in diabetic rats [18]. The studies evaluating γ-oryzanol antioxidant and potential effects in the control of diabetes are compiled in Table 4, Table 5, Table 6 and Table 7.

#### 4.5.2. Phytic Acid

Despite being considered an anti-nutrient, due to the formation of complexes (with calcium and magnesium) during the digestion process, PA has been studied for its hypolipidemic and antioxidant effects, the latter mainly through the suppression of oxidative reactions catalyzed by Fe [28]. PA benefits for diabetes are related to the prevention of hyperglycemia through the reduction of starch digestibility [135,136]. In fact, animal studies show the reduction of blood glucose when PA is added to diet [11,95,96,97,98,137,138]. It is also thought that PA reduces starch digestion rate, binding to the salivary and pancreatic amylase enzyme through phosphate bonds or by binding with minerals that catalyze the activity of amylase, such as calcium, restricting its activity [97]. The studies evaluating the potential effects of PA in the control of diabetes are compiled in Table 4, Table 5, Table 6 and Table 7.

#### 4.5.3. Ferulic Acid

The antioxidant capacity of FA seems to be related to the presence of electron donor groups in the benzene ring, allowing FA to end the chain reactions of free radicals, and to ROS elimination [129,139] by increasing the activity of antioxidant enzymes [17,133]. FA also stands out for its effects in diabetes control [140] at various levels: reducing blood glucose through inhibition of α-glucosidase [95] and improving insulin secretion in diabetic rat models [141] and in pancreatic cells [142]. The main FA effects towards diabetes are summarized in Table 4, Table 5, Table 6 and Table 7.

#### 4.5.4. γ-Aminobutyric Acid (GABA)

GABA is a neurotransmitter found in the central nervous system with two important functions: a tranquilizer and a hypotensive [78,80]. Protective actions of GABA have been associated with the prevention of Alzheimer′s disease [143], reduction of blood pressure [144], and prevention of diabetes [107]. As for its antidiabetic properties, GABA is known to control insulin secretion by pancreatic β-cells [116] through glucagon inhibition. In addition, GABA inhibits glucagon secretion by promoting the conversion of α-cells into β-cells, thereby contributing to the increase of β-cell mass and insulin production [116,145,146]. Overall, studies have shown that regular consumption of GABA-rich foods can contribute to the increase of insulin secretion and decrease blood glucose [100,108]. The studies evaluating GABA’s potential effects in the control of diabetes are compiled in Table 4, Table 5, Table 6 and Table 7.

#### 4.5.5. Tocopherols and Tocotrienols (Vitamin E)

The potential effect of vitamin E in diabetes control is related to its antioxidant properties against ROS and nitrous compounds affecting β-cell function [147,148,149]. Diet supplementation for 3 months with vitamin E decreased plasma glucose and peroxide levels and improved insulin levels in overweight individuals [101]. Vitamin E extracts also showed beneficial effects for reducing the blood glucose levels of diabetic rats [102]. These fat-soluble compounds can decrease insulin resistance and improve glucose metabolism in diabetic rats by modulation of peroxisome proliferator-activated receptors and regulation of energy metabolism [150]. The studies evaluating the potential effects of tocopherols and tocotrienols in the control of diabetes are compiled in Table 4 and Table 7.

## 5. Conclusions

Rice components, especially macronutrients (starch, proteins) and the compounds present in bran (γ-oryzanol, phytic acid, ferulic acid, γ-aminobutyric acid, tocopherols, and tocotrienols) have been pointed to as having potential bioactive effects that could be beneficial for diabetes prevention and control. The protective effects of rice bran constituents are mostly related to the reduction of food GI, partially due to the inhibition of α-amylase and α-glucosidase (vitamin E is an exception), with implications in the reduction of blood glucose. Ferulic acid stands out as having bioactive effects towards several diabetes pathological processes, including the potential inhibition of protein aggregation and amyloidogenesis. Phytic acid, ferulic acid, and rice proteins (albumin) are suggested to modulate the activity of glucose transporters, whereas γ-oryzanol, γ-aminobutyric acid, tocopherol, and tocotrienol bioactivities are associated with the attenuation of β-cell dysfunction in the pancreas. Despite evidences of the positive effects of rice macronutrients and bran bioactive compounds on diabetes, further studies are needed to support the exploitation of rice compounds in diabetes control.

## Figures and Tables

**Figure 1 foods-10-01992-f001:**
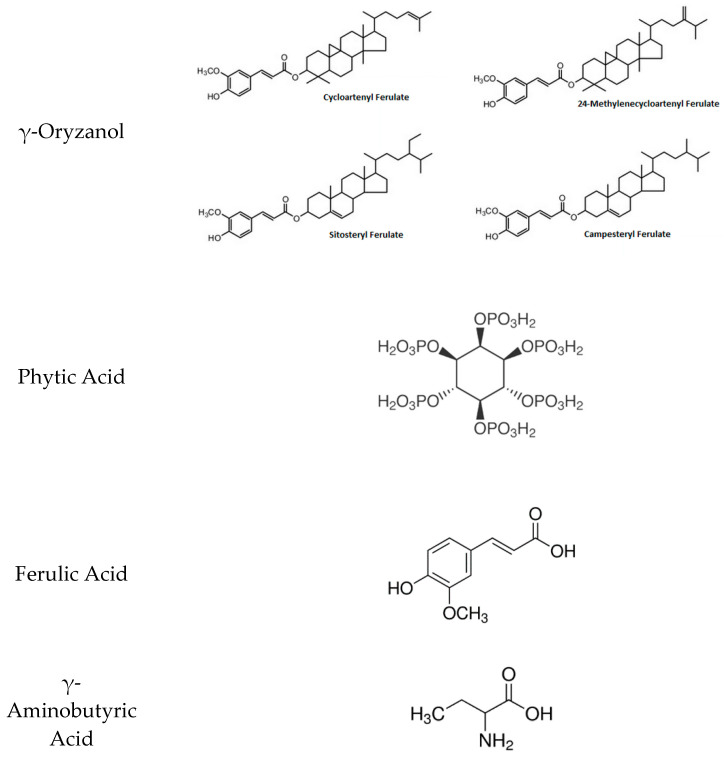
Chemical structure of rice bran compounds.

**Table 1 foods-10-01992-t001:** Basic composition of white rice and rice bran (%) ^1^.

Nutrients	White Rice	Brown Rice	Rice Bran
Carbohydrates Starch	77.0–89.0 77.6	73.0–87.0 66.4	34.0–62.0 13.8
Lipids	0.3–0.5	1.6–2.8	15.0–19.7
Protein	4.5–10.5	4.3–18.2	11.3–14.9
Crude Fiber Dietary Fiber	0.2–0.5 0.7–2.7	0.6–1.0 2.9–4.4 *	7.0–11.4 19.0–29.0 *
Ash	0.3–0.8	1.0–1.5	6.6–9.9

^1^ Adapted from [7] and * [8]; Published by AACC International Press.

**Table 2 foods-10-01992-t002:** Main bioactive compounds in white rice and rice bran (mg/100 g).

Compounds	White Rice	References	Rice Bran	References
γ-oryzanol	5.0–7.2	[20]	59.4–912.0	[21,22,23,24]
Phytic acid	13.0–1104.0	[25,26,27]	4000.0–22,500.0	[25,26,28,29]
Ferulic acid	0.5–8.4	[30,31]	1.4–225.4	[31]
γ-Aminobutyric acid	0.3–0.7	[32]	10.7–58.0 (before grain germination) 90.0–350.0 (after grain germination)	[33,34,35]
Tocopherols and tocotrienols (vitamin E)	0.6–1.8	[20]	17.0–22.9	[20,36,37]

**Table 3 foods-10-01992-t003:** Glycemic index (GI) values of different varieties of cooked rice according to ISO 26642:2010 ^1^.

Type of Rice	GI
Glutinous white rice (unique study)	98
Glutinous white rice (unique study)	94
Sticky rice, Thai glutinous rice (unique study)	92
Jasmine white rice (mean of 18 studies)	89
Japanese Style Sushi white rice (unique study)	85
Arborio risotto rice (unique study)	69
Carnaroli white rice (unique study)	64
Parboiled rice (mean of 10 studies)	64
Rice Long grain (mean of 6 studies)	62
Basmati white rice boiled (mean of 10 studies)	60
Taiken brown rice (japonica rice) (unique study)	58
Bapatla brown rice (indian rice) (unique study)	58

^1^ Adapted from [40]; Published by the American Society for Nutrition.

**Table 4 foods-10-01992-t004:** Studies on the effect of macronutrients and bioactive rice compounds in the glycemic index of foods, blood glucose and insulin, and oxidative stress.

Compound	Model/Dose	Effect *	Reference
Rice Proteins	Wistar rats, diet supplemented with 50–200 mg rice albumin/Kg weight after 15 min of oral starch or glucose administration (1 g/Kg weight)	↓ Blood glucose	[93]
	In vitro digestion: native rice protein (12%, pressure cooking 95 °C 30 min) + wheat starch (70%) + digestive enzymes, during 30 min	↓ RDS content after pressure cooking Native rice protein promotes starch-protein interaction and restricts starch hydration and enzymatic cleavage	[45]
	Chinese males, diet: drink rich in carbohydrates (50 g) + rice protein (24 g) during 15, 30, 45, 60, 90, 120, 150, and 180 min	↓ Post-prandial blood glucose	[94]
	In vitro digestion: rice flour with and without endogenous protein (8.4%) + starch enzymes, during digestion	↓ Starch digestibility in rice flour with endogenous protein (GI = 92.3 to GI = 88.9)	[55]
Resistant Starch	In vitro digestion: cookies with 50% rice flour + 50% RS from de-branched (RSa) or from acid and heat-moisture (RSc)	↓ Starch digestibility reducing cookies estimated GI	[42]
	In vivo: healthy and T2DM individuals: meals contained PPB-R-203-derived rice/noodles enriched with RS (10% of starch),3 days	↓ Blood glucose and insulin ↓ Postprandial hyperglycemia in T2DM individuals	[44]
γ-oryzanol	STZ treated Wistar rats; diet supplemented with oryzanol from rice bran 50 and 100 mg/Kg/day, 8 weeks	↓ Serum glucose ↓ Oxidative stress	[18]
Phytic acid (PA)	Wistar rat diet supplemented with PA at 10 to 13 g diet/day, 3 weeks	↓ Blood glucose	[95]
	Diabetic Wistar albino rats supplemented with 650 mg PA/kg, 28 days	↓ Blood glucose	[96]
	C57BL/6N mice; 3 g diet/day contained 0.5% PA, 7 weeks	↓ Blood glucose	[97]
	Diabetic KK mice, diet with 0.5–1.0% sodium phytate, 8 weeks	↓ Blood glucose	[98]
Ferulic Acid (FA)	C57BL/KsJ-db/db diabetic mice; diet supplemented with FA from rice, 0.05 g/kg/day, 17 days	↓ Blood glucose ↑ Plasma insulin	[99]
γ-Aminobutyric acid (GABA)	C57BL/6 mouse diet supplemented with 2 mg/mL GABA, 20 weeks	↓ Glucose intolerance ↓ Fasting blood glucose	[100]
Tocopherols and tocotrienols (vitamin E)	Overweight individuals group supplemented with 800–1200 IU vitamin E/day, 3 months	↓ Fasting plasma glucose and insulin	[101]
	Sprague–Dawley diabetic rats; diet with vitamin E extract (1 g/kg weight), 12 weeks	↓ Blood glucose	[102]

* “↑” means increase; “↓” means decrease.

**Table 6 foods-10-01992-t006:** Effect of macronutrients and bioactive rice compounds in the modulation of SGLT1 and GLUT2.

Compound	Model/Dose	Effect	Reference
Rice Proteins	Intestinal STC-1 cells with 100 μg of tripsin-digested rice albumin/mL, 48 h	Suppressed SGLT1	[113]
Phytic acid (PA)	Piglets (Yorkshire-Landracex Duroc) supplemented with 2 g PA or Na phytate/1 kg diet, 10 days	↓ Crypt depth in the jejunum ↓ SGLT1 expression in the duodenum, jejunum, and ileum ↓ Nutrient utilization in pigs, which is involved in glucose and Na absorption	[114]
Ferulic acid (FA)	Caco-2 cells, 0–0.1 mg/mL FA, 15 min; *Xenopus laevis* oocytes, 100–300 µM FA, 30 min	↓ Glucose uptake in Caco-2 cells Block glucose uptake in oocytes ≥100 μM ↓ GLUT2 in oocytes	[115]

“↑” means increase; “↓” means decrease.

**Table 7 foods-10-01992-t007:** Effect of bioactive rice compounds against pathological mechanisms associated with diabetes.

Source	Model/Dose	Effect	Reference
γ-oryzanol	C57BL/6J mouse, diet supplemented with oryzanol 320 μg/g weight/day, 13 weeks	↓ ER stress in pancreatic β-cells ↓ Pancreatic islet dysfunction ↑ Protection of β-cells against apoptosis	[118]
Ferulic acid (FA)	Cell-free, human amylin peptide + 10–40 µM FA, 192 h	↓ IAPP amyloid formation by 27.7% to 22.6%	[134]
γ-Aminobutyric acid (GABA)	CD1 mice + 2 injections of 20 μmol GABA/mouse during 48 h; INS-1 cells challenged with streptozotocin (STZ) (15 mM, 24 h) + 1, 10, 100 μM of GABA	↑ Islet cell function ↑ Protection of β-cell from apoptosis	[116]
	CD1 or C57 mouse, supplemented with 6 mg GABA/mL, 10 weeks	↑ β-cell proliferation ↑ Insulin secretion	[117]
Tocopherols and tocotrienols (vitamin E)	C3H/AnLCS^a^CS^a^ mouse induced with alloxan and supplemented with 50 mg α-tocopherol/100 g diet, 14 weeks	↑ Insulin secretion ↓ Apoptosis caused by oxidative stress	[19]

“↑” means increase; “↓” means decrease.

**Table 5 foods-10-01992-t005:** Effect of macronutrients and bioactive rice compounds in α-amylase and α-glucosidase activities.

Compound	Model/Dose	Effect	Reference
Rice Proteins	Protein hydrolysates from rice bran (cultivar Reiziq); 10 mg/mL + in vitro enzyme preparations + 1% starch solution, 240 min	High inhibition of starch enzymes activities by albumin and glutelin	[65]
γ-oryzanol	40 µL α-amylase 50 U/mg + 0.5 mL γ-oryzanol + 40 µL of starch 1%; 100 µL α-glucosidase 50 U/mg + 5 mg γ-oryzanol, 30 min	Inhibition of α-amylase (IC_50_ = 0.78 mg/mL) and α-glucosidase (IC_50_ = 0.81 mg/mL) *	[106]
Phytic acid (PA)	1 U/mL α-amylase + 0.25–8 µg PA/mL, 3 min; 5–100 µ g PA/mL + ~0.1 mL α-glucosidase + starch solution 0.5%, 60 min	Inhibition of α-amylase (IC_50_ = 1.2 µg/mL) and α-glucosidase (IC_50_ =3.2 µg/mL) *	[96]
Ferulic Acid (FA)	In vitro: 100 mg Rat intestinal acetone powder + 10 µL maltase or 40 µL sucrase + 0–1 mM FA, 30–60 min	Inhibition of α-glucosidase (IC_50_ = 0.79 mM) and intestinal maltase and sucrase (IC_50_ = 0.45 mM) *	[107]
	20 µL α-glucosidase from baker’s yeast + 20 µL FA, 30 min	Inhibition of α-glucosidase (IC_50_ = 0.8 mg/mL) *	[99]
γ-Aminobutyric acid (GABA)	50 µL α-glucosidase 0.2 U/mL + 50 µL cell-free extract from yeasts containing GABA (86.2–179.2 µL/mL), 30 min	Anti-hyperglycemic effect High inhibition of α-glucosidase (up to 72.3%)	[108]

* IC_50_-concentration at which a compound exerts half of its maximal inhibitory effect.

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
