# Peer review of "Rice Compounds with Impact on Diabetes Control"

_foods, 2021, doi:10.3390/foods10091992_

Round 1
Reviewer 1 Report
The authors of the manuscript pay attention to the health benefits of rice products other than the most commonly used white rice. In the article, a lot of attention was paid to influence of rice bran on lipid metabolism and glicemic control, however, that there was little information on the nutritional and health-promoting value of brown rice or parbiold rice. It is worth supplementing this information.
It is worth adding a summary of information (e.g. in a table) about the GI of various rice products and meals with their addition, as well as information about the glycemic load of these products
As the authors themselves write, the assessment of carbohydrate digestion in vitro may indicate the rate of starch hydrolysis and availability of simple sugars from the product / meal, but it is not a glycemic index and it should not be called that. The only way to indicate the effect of carbohydrate products on postprandial glycemia is in vivo IG determination.
Most of the cited works on the influence of phytic acid, GABA, and tocols on the carbohydrate metabolism and oxidative stress does not directly relate to the influence of rice products, it is worth mentioning it in the text. The method of supply of the described bioactive ingredients and their doses used in model animal experiments may not be adequate to their effect in humans when consumed as an ingredient of a meal with rice products.
Author Response
Response to Reviewer 1 Comments
Dear Editors,
The authors are very grateful to the reviewers for the comments to improve the scientific quality of the manuscript. The reviewers' suggestions are highly appreciated, and the manuscript has been changed improved according to the reviewers’ remarques.
Reviewer 1:
Point 1: The authors of the manuscript pay attention to the health benefits of rice products other than the most commonly used white rice. In the article, a lot of attention was paid to influence of rice bran on lipid metabolism and glycemic control, however, that there was little information on the nutritional and health-promoting value of brown rice or parbiold rice. It is worth supplementing this information.
Response 1: The manuscript was complemented with the requested nutritional information on brown rice. It is now possible to compare its nutritional value with white rice and rice bran in table 1. To further complement this information, some studies on the benefits of brown rice-supplemented diets, namely on the effect on the glycemic index, were highlighted as follows (Lines 50 to 54): “Brown rice has a higher content of nutritional compounds compared with white rice, except starch (Table 1). The health benefits of brown rice are mainly due to their proteins, lipids, vitamins, minerals and dietary fiber contents [7,8]. Several in vivo studies show the benefits of brown rice-based or brown rice-supplemented diets in diabetes control, especially in reducing the glycemic index [9-11]”.
Point 2: It is worth adding a summary of information (e.g. in a table) about the GI of various rice products and meals with their addition, as well as information about the glycemic load of these products.
Response 2: A table was included showing glycemic index of major rice types and rice varieties as determined in vivo (Lines 208 to 209).
Point 3: As the authors themselves write, the assessment of carbohydrate digestion in vitro may indicate the rate of starch hydrolysis and availability of simple sugars from the product / meal, but it is not a glycemic index and it should not be called that. The only way to indicate the effect of carbohydrate products on postprandial glycemia is in vivo IG determination.
Response 3: The authors thank the correction and explanation. The definition of glycemic index has been reformulated to indicate that it is determined by in vivo methods. The in vitro methods of starch digestion were redefined as only estimation and expectation methods (Lines 527 to 540).
Point 4: Most of the cited works on the influence of phytic acid, GABA, and tocols on the carbohydrate metabolism and oxidative stress does not directly relate to the influence of rice products, it is worth mentioning it in the text. The method of supply of the described bioactive ingredients and their doses used in model animal experiments may not be adequate to their effect in humans when consumed as an ingredient of a meal with rice products.
Response 4: The authors thank for the observation. To clarify this issue, we added the following explanation preceding table 4. (Lines 547 to 552): “It is important to note that these studies evaluated the effects of pure forms of rice bioactive compounds or as supplements in other foods or meals, and mostly carried out on animals. Thus, the potential benefits for human health should be extrapolated with caution”

Reviewer 2 Report
The work submitted for review has scientific values and a good starting point for further researches. The structure of paper follows the concentrations of components, giving an overview about their ranges and describing their effects on diabetic control comprehensively, giving jumping points to the references with detailed description. In my opinion the manuscript need language verification in the submitted version
Author Response
Response to Reviewer 2 Comments
Reviewer 2
Point 1: The work submitted for review has scientific values and a good starting point for further researches. The structure of paper follows the concentrations of components, giving an overview about their ranges and describing their effects on diabetic control comprehensively, giving jumping points to the references with detailed description. In my opinion the manuscript need language verification in the submitted version.
Response 1: The authors thank the reviewer for the careful revision of the manuscript. The revised version of the manuscript was verified using a spelling and grammar checker.
